# The Chx10-Traf3 Knockout Mouse as a Viable Model to Study Neuronal Immune Regulation

**DOI:** 10.3390/cells10082068

**Published:** 2021-08-12

**Authors:** Jami M. Gurley, Grzegorz B. Gmyrek, Elizabeth A. Hargis, Gail A. Bishop, Daniel J. J. Carr, Michael H. Elliott

**Affiliations:** 1Department of Ophthalmology, Dean McGee Eye Institute, University of Oklahoma Health Sciences Center (OUHSC), 608 Stanton L. Young Blvd., Oklahoma City, OK 73104, USA; grzegorz-gmyrek@ouhsc.edu (G.B.G.); elizabeth-hargis@ouhsc.edu (E.A.H.); dan-carr@ouhsc.edu (D.J.J.C.); michael-elliott@ouhsc.edu (M.H.E.); 2Department of Microbiology and Immunology, University of Iowa and VAMC, Iowa City, IA 52242, USA; gail-bishop@uiowa.edu; 3Department of Microbiology and Immunology, University of Oklahoma Health Sciences Center (OUHSC), 608 Stanton L. Young Blvd., Oklahoma City, OK 73104, USA

**Keywords:** neural retina, Traf3, immunity, inflammation, neurodegeneration, central nervous system, vision, lipopolysaccharide

## Abstract

Uncontrolled inflammation is associated with neurodegenerative conditions in central nervous system tissues, including the retina and brain. We previously found that the neural retina (NR) plays an important role in retinal immunity. Tumor necrosis factor Receptor-Associated Factor 3 (TRAF3) is a known immune regulator expressed in the retina; however, whether TRAF3 regulates retinal immunity is unknown. We have generated the first conditional NR-*Traf3* knockout mouse model (*Chx10*-Cre/*Traf3*^f/f^) to enable studies of neuronal TRAF3 function. Here, we evaluated NR-*Traf3* depletion effects on whole retinal TRAF3 protein expression, visual acuity, and retinal structure and function. Additionally, to determine if NR-*Traf3* plays a role in retinal immune regulation, we used flow cytometry to assess immune cell infiltration following acute local lipopolysaccharide (LPS) administration. Our results show that TRAF3 protein is highly expressed in the NR and establish that NR-*Traf3* depletion does not affect basal retinal structure or function. Importantly, NR-*Traf3* promoted LPS-stimulated retinal immune infiltration. Thus, our findings propose NR-*Traf3* as a positive regulator of retinal immunity. Further, the NR-*Traf3* mouse provides a tool for investigations of neuronal TRAF3 as a novel potential target for therapeutic interventions aimed at suppressing retinal inflammatory disease and may also inform treatment approaches for inflammatory neurodegenerative brain conditions.

## 1. Introduction

Immune responses have evolved to defend host tissues from foreign invaders. These responses are particularly important for maintenance of post-mitotic central nervous system (CNS) neurons in the brain and retina following infection or injury, as neuronal cells lack the ability to regenerate after cell death. However, prolonged overactivation of these immune processes can result in neurotoxicity, which commonly occurs during inflammatory neurodegenerative disease progression and ultimately results in neuronal cell death. Unfortunately, neuroinflammatory regulation is not well understood and is further confounded by the complex nature of immune privilege mechanisms present in neuronal tissues [1]. Often considered a “window to the brain”, the retina is an accessible extension of the CNS that is more amenable to less-invasive in vivo observations [2,3].

Thus, in addition to advancing our knowledge regarding blindness-causing neurodegenerative diseases, retinal mouse models also have the potential to inform neurodegenerative brain research.

The retina functions to receive, convert, and transmit light information in the form of electrical and chemical signals that are interpreted by the brain for visual processing [4]. Retinal neurons, including photoreceptor (i.e., rods and cones), horizontal, bipolar, amacrine, and ganglion cells participate in a signaling cascade (i.e., phototransduction) that transmits this translated information to the brain for further visual interpretation [5,6]. Chronic retinal neurodegeneration, thus, leads to vision loss and, eventually, complete blindness [7,8]. In addition to the aforementioned neuronal populations, many other cell types are present in the retina and support healthy neuronal function, including providing immune protection in response to infection and/or injury. Microglia, astrocytes, and retinal pigment epithelia (RPE) have long been known to play important roles in retinal immune homeostasis, and cells of the retinal vasculature modulate the highly-selective permeability of the blood–retina barrier (BRB) during inflammation [9,10,11,12]. Recently, we found that the neural retinal (NR) compartment, which is comprised of all retinal neurons and Müller glial cells, also plays an important role in retinal immune regulation [13].

In a previous study, we identified Tumor necrosis factor Receptor-Associated Factor 3 (TRAF3) as a novel potential immune regulator associated with retinal membrane fractions [13]. TRAF3 has been well established as a cytosolic and nuclear adapter protein that directly interacts with cell surface receptors to regulate a wide variety of signaling pathways and cellular processes [14,15,16,17,18]. Global-*Traf3* depletion in mice results in neonatal lethality [19]. However, conditional-*Traf3* knockout mouse models have been successfully generated and, in conjunction with cell culture models, have provided valuable insights regarding additional cell-specific TRAF3 functions, including cell survival, metabolic regulation, growth, proliferation, and immune modulation [20,21,22,23,24,25,26]. Notably, gene and protein databases report high *Traf3* expression in the brain and retina [27]. However, studies regarding TRAF3 function in neuronal tissues have only recently begun to be investigated [24,28,29]. Further, to our knowledge, no retinal-specific mouse models have yet been developed and retinal TRAF3 function has not been explored. Thus, the goals for this study were (1) to develop a novel genetic NR-*Traf3* depletion model, (2) to investigate potential *basal* effects of NR-*Traf3* ablation on retinal tissue architecture and function, and (3) to determine if NR-*Traf3* plays a functional role in retinal immunity. Our study, thus, also evaluates the NR-*Traf3* mouse as a useful model for assessing in vivo neuronal-specific TRAF3 function.

## 2. Materials and Methods

### 2.1. Materials

Information for the proceeding reagents and materials is as follows: n-octyl-β-D-glucopyranoside (Cat# 494459), Triton X-100 (Cat# X100), NaCl (Cat# S3014), EDTA (Cat# 324506), complete mini protease inhibitor cocktail tablets (Cat# 11836170001), bovine serum albumin (BSA; Cat# A7906), glycerol (Cat# G5516), and lipopolysaccharides from *Salmonella enterica* serotype typhimurium (LPS; Cat# L2262) were purchased from MilliporeSigma (Burlington, MA, USA). The Pierce BCA protein assay kit (Ref 23227), Invitrogen (Waltham, MA, USA) Novex WedgeWell 8% Tris-Glycine Gels (1.0 mm × 12 well; Ref XP00082BOX), BenchMark Prestained Protein Ladder (Ref 10748-010), XCell SureLock Mini-Cell gel electrophoresis system (Ref EI0001), 10× Novex Tris-Glycine SDS Running Buffer (Cat# LC2675-5), and SuperSignal West Dura Extended Subtrate (Ref 34076) were purchased from ThermoFisher Scientific (Waltham, MA, USA). Nitrocellulose membrane (0.45 µm; Cat# 1620115), Mini Trans-Blot Cell system (Cat# 1703930), and PowerPac Basic power supply (Cat# 1645050) were from BioRad (Hercules, CA, USA). Normal horse serum blocking solution (S-2000-20) was purchased from Vector Laboratories (Burlingame, CA, USA).

Antibodies for Western blotting were obtained as follows: primary rabbit polyclonal anti-TRAF3 was purchased from Novus Biologicals (#NBP1-86639; Littleton, CO, USA). Primary mouse monoclonal anti-Actin was purchased from ThermoFisher (#MA1-744). Donkey anti-rabbit (NA934V) and sheep anti-mouse (NA931V) horseradish peroxidase (HRP)-linked secondary antibodies were purchased from Global Life Sciences Solutions (Marlborough, MA, USA). Antibodies for retinal immunofluorescence were obtained as follows: primary mouse monoclonal anti-Glutamine Synthetase was purchased from MilliporeSigma (#MAB302, clone GS-6). Primary rat monoclonal mouse anti-CD31 was purchased from Dianova (#DIA-310; Hamburg, Germany). Alexa Fluor 488-conjugated goat anti-mouse secondary antibody (A-11001) was purchased from ThermoFisher. Alexa Fluor 647-conjugated donkey anti-rat secondary antibody (Code: 712-605-150) was purchased from Jackson ImmunoResearch (West Grove, PA, USA). Antibodies for flow cytometry were obtained as follows: Pacific Blue-conjugated anti-mouse CD45 (clone 30-F11), PE-conjugated anti-mouse GR1 (Ly6G/Ly6C; clone RB6-8C5), Spark Blue 550-conjugated anti-mouse CD45 (clone 30-F11), APC-Cy7-conjugated anti-mouse Ly6C (clone HK1.4), PerCP-Cy5.5-conjugated anti-mouse Ly6G (clone 1A8), PE-conjugated anti-mouse/human CD11b (clone M1/70), PE-Cy7-conjugated anti-mouse CD115 (CSF-1R; clone AFS98), APC-conjugated anti-mouse F4/80 (clone BM8), BV650-conjugated CCR2 (CD192; clone SA203G11), and BV605-conjugated MHC II (clone M5/114.15.2) were all purchased from Biolegend, San Diego, CA, USA. The Zombie Aqua fixable viability reagent for 7-color flow cytometry was also purchased from Biolegend, San Diego, CA, USA.

### 2.2. Chx10-Traf3 KO Model

All animal procedures were approved by the University of Oklahoma Health Sciences Center Institutional Animal Care and Use Committee (OUHSC IACUC) and follow the Association for Research in Vision and Ophthalmology (ARVO) Statement for the Use of Animals in Ophthalmic and Visual Research. Neural retinal-specific *Traf3* knockout (NR-*Traf3* KO) animals were generated via Cre-Lox technology by breeding mice carrying *Chx10*-promoter-driven Cre recombinase with *Traf3*-floxed mice provided by G.A. Bishop (Figure 1a) [20]. Expression of Chx10-Cre (stock#: 005105, The Jackson Laboratory, Bar Harbor, ME) occurs specifically in neural retinal progenitor cells, resulting in recombination in retinal neurons (photoreceptor, bipolar, and ganglion cells), and Müller glial cells (Figure 1b) [13,30,31]. Recombination within the *Traf3* gene occurs via Lox P sites located upstream of exon 1 and downstream of exon 2 [20].

### 2.3. Whole Retinal Tissue Lysis

Whole retinal protein extracts were obtained via brief sonication of one retina (~6–7 μg) per sample in 65 μL 2× lysis buffer (120 mM octyl glucoside, 2% Triton X-100, 20 mM Tris-HCl, pH 7.4, 200 mM NaCl, 1 mM EDTA) containing a protease inhibitor cocktail tablet (Roche). Lysates were cleared by 10-min centrifugation at 13 K rpm, at 4 °C. Protein concentration was determined via BCA assay, using bovine serum albumin standard, and samples were diluted in 6× Laemmli buffer and stored at −80 °C until further use.

### 2.4. Western Blotting

Whole retinal tissue protein separation was achieved via SDS-PAGE. Proteins were then transferred to nitrocellulose for immunoblotting with the following primary antibodies and dilutions: rabbit polyclonal anti-TRAF3 (1:500; Novus Biologicals, Cat# NBP1-86639), mouse monoclonal anti-Actin (1:1000; ThermoFisher, #MA1-744). Immunoreactivity was detected using species-appropriate horseradish peroxidase (HRP)-conjugated secondary antibodies (1:5000 for TRAF3 and 1:10,000 for Actin; GE Healthcare, Cleveland, OH, USA). Western blots were imaged via HRP chemiluminescent detection (Azure Biosystems, Inc.; Dublin, CA, USA) and densitometric analyses were performed using Image Studio Lite software (LI-COR Biosciences; Lincoln, NE, USA).

### 2.5. Immunohistochemistry and Confocal Microscopy

After adult mice were euthanized, whole eye globes were fixed in Prefer fixative (Anatech, Ltd., Battlefield, MI, USA), embedded in paraffin, and cut into 5 μm sections. Deparaffinization was performed via successive incubations using the following protocol: 3× for 5 min each in fresh xylene, 2× for 3 min each in 100% ethanol, and 1× for 3 min in 95% ethanol. Sections were then rinsed with deionized water 3×, were permeabilized with 1% Triton X-100 in PBS, then incubated with blocking solution (10% normal horse serum, 0.1% Triton X-100, in 1XPBS) for one hour prior to immunohistochemistry, which was performed as previously described using either hematoxylin and eosin (H&E) staining or the following antibodies: primary mouse monoclonal anti-Glutamine Synthetase (1:5000, MilliporeSigma #MAB302, clone GS-6) and rat monoclonal anti-mouse CD31 (1:100, Dianova #DIA-310). Immunoreactivity was detected with species-appropriate Alexa Fluor-conjugated secondary antibodies (1:500; goat anti-mouse 488, donkey anti-rat 647). Images of immunostained retinal tissue sections were captured using an Olympus FV1200 laser scanning confocal microscope and visualized using FluoView software (Olympus). Pseudocolors were assigned to images as follows: Glutamine Synthetase, green; CD31, red.

### 2.6. Optokinetic Tracking (OKT)

Visual acuity was assessed via non-invasive observations of unrestrained mouse behavioral (head turning) responses to a rotating visual gradient as described by Prusky et al. [32]. Briefly, systematic increases in spatial frequencies were used to determine maximum visual thresholds for both left and right eyes in all mice (OptoMotry; CerebralMechanics, Lethbridge, AB, Canada). For each mouse, two maximum threshold measurements per eye were recorded and averaged.

### 2.7. Optical Coherence Technology (OCT)

Spectral domain optical coherence technology/tomography (SD-OCT; Bioptigen; Durham, NC, USA), in conjunction with InVivoVue Diver software (Version 2.4) analysis, was used for in vivo assessment of retinal structure. After determining that no apparent anomalous retinal defects were present, quantitative retinal layer thickness measurements for each animal were calculated individually for the right eye (RE) and left (LE) eye of each animal. Automated total and layer-specific retinal thickness values were calculated via manual determination of 10 retinal boundaries, which defined 9 distinct retinal tissue layers. These retinal tissue layers were obtained for 8 “inner” and “outer” retinal zones (relative to the central optic nerve head (ONH)) comprising Superior, Temporal, Inferior, and Nasal positions of each retina. Mirrored retinal zone measurements for the left and right eyes of each animal were then averaged to obtain a single measurement, per animal, for each retinal zone location. Following determination of averaged individual mouse retinal layer measurements, group data were then combined to assess potential genotype-specific differences in total retinal thickness as well as for each of the smaller encapsulated subdivisions and individual retinal layers.

### 2.8. Electroretinography (ERG)

ERGs were recorded as described previously [33,34]. Briefly, overnight dark-adapted mice were anesthetized with ketamine (100 mg/kg) and xylazine (10 mg/kg), pupils were dilated with 0.5% atropine and 2.5% phenylephrine, gold wire electrodes were positioned on corneas, a reference electrode was placed inside the mouth, and a ground electrode was places in the tail. Rod-driven responses were assessed by presenting increasing scotopic stimuli (−3.7 to 2.6 log scotopic candela (cd) × s/m^2^) via a Colordome Espion ERG recording system (Diagnosys, Lowell, MA, USA). Intensity response relationships for a- and b-wave were fit using the GraphPad Prism Michaelis–Menten equation in order to calculate physiological maximum amplitudes (i.e., Vmax) [33,34].

### 2.9. Endotoxin-Induced Uveitis Model

Intravitreal injection was used to locally administer 1 µg LPS (*Salmonella typhimurium*; Sigma) in order to induce ocular inflammation as described previously [13]. Briefly, mice anesthetized with intraperitoneal ketamine (100 mg/kg)/xylazine (10 mg/kg) injection received 1 µL contralateral injections of either saline (1× PBS) vehicle or LPS (diluted in 1X PBS vehicle), via a 10 µL glass syringe (Hamilton) with a 33-gauge needle, into the vitreous chamber of each eye. To ensure minimal damage to ocular tissues, the eye was gently proptosed and held in place by clasping the surrounding eyelids during injections. Whole retinal tissue was harvested, using the “Winkling” method, 24 h after immune induction for downstream assessment of innate immune infiltration using flow cytometry [35,36,37].

### 2.10. Flow Cytometry

Processing and staining of retina tissue for flow cytometry was performed as previously described, but with some minor modifications [13]. Briefly, following whole animal perfusion, extracted retinal tissue was minced and digested with liberase TL for 20 min at 37 °C. The cell suspension from digested tissue was filtered through 40 µm cell strainer and washed with PBS supplemented with 2% FCS and 2 mM EDTA (called herein as Staining Buffer or SB). Fc receptors were blocked using anti-mouse CD16/32 (Fc-block; Invitrogen) at 4 °C for 10 min. For 3-color flow cytometry, cells were stained with the following antibodies: Pacific Blue-conjugated anti-mouse CD45 (clone 30-F11), PE-conjugated anti-mouse GR1 (Ly6G/Ly6C; clone RB6-8C5) and APC-conjugated anti-mouse F4/80 (clone BM8). For 7-color flow cytometry, cells were first stained with cell viability dye (Zombie Aqua) for 15 min at room temperature followed by washing with SB. Finally, the cells were Fc-blocked (10 min at 4 °C) and stained (25 min on ice) with anti-mouse antibody cocktail containing Spark Blue 550-conjugated anti-mouse CD45 (clone 30-F11), APC-Cy7-conjugated anti-mouse Ly6C (clone HK1.4), PerCP-Cy5.5-conjugated anti-mouse Ly6G (clone 1A8), PE-conjugated anti-mouse/human CD11b (clone M1/70), PE-Cy7-conjugated anti-mouse CD115 (CSF-1R; clone AFS98), APC-conjugated anti-mouse F4/80 (clone BM8), BV650-conjugated CCR2 (CD192; clone SA203G11), and BV605-conjugated MHC II (clone M5/114.15.2). For 3-color flow cytometry analysis, data were acquired using a MacsQuant flow cytometer (Miltenyi Biotec, Auburn, CA, USA). Events were gated using forward and side scatter, along with sequential gating to distinguish infiltrating cells as previously described [38]. For 7-color flow cytometry analysis, sample data were acquired using 4-laser spectral flow cytometer Aurora (Cytek Biosciences, Fremont, CA, USA) containing 16 violet, 14 blue, 10 yellow-green and 8 red channels (4L-16V-14B-10YG-8R). Each fluorochrome peaked in a separate channel and a general spectra pattern was validated based on the references provided by Cytek Biosciences (online resources: Cytek Full Spectrum Viewer; website: https://spectrum.cytekbio.com/ (accessed on 11 March 2021). Compensation for spectral unmixing was performed using reference controls and unmixing wizard integrated in SpectroFlo software. The gating strategy was established based on running FMO (Fluorescence Minus One) retina samples. Acquired data for all samples were exported as FCS files and analyzed with FlowJo software version 10.7.1 (BD Biosciences, Ashland, OR, USA) as previously described [13].

### 2.11. Other Methods

Numerical data are represented as the mean ± standard error of the mean (SEM). Statistically significant differences among groups were determined using GraphPad Prism (Version 8.1.2) Student’s *t*-test, one-way analysis of variance (ANOVA), or two-way ANOVA where appropriate and are described in corresponding figure legends. Statistically significant data were defined as having a *p*-value < 0.05. For functional studies, *Chx10*-Cre^+^ control and *Traf3*^flox/flox^ controls were pooled as “NR-Traf3 WT,” as we did not observe any differences between these groups. For Western blot data, *Chx10*-Cre^−^/*Traf3*^flox/flox^ animals were used as controls.

## 3. Results

### 3.1. The Majority of Retinal TRAF3 Protein Expression Is in the Neural Retinal (NR) Compartment

We and others have previously demonstrated that the *Chx10*-Cre/Lox genetic knockout model is an effective tool for targeted neural retinal-specific gene depletion [13,30,31]. Here, we generated conditional neural retinal-*Traf3* knockout mice (NR-*Traf3* KO) by breeding our *Chx10*-Cre mouse line with a *Traf3*^flox/flox^ mouse line (Figure 1a). This NR-*Traf3* KO is characterized by *Cre*-mediated recombination of *Traf3* exons 1 and 2. As *Chx10* (also known as Ceh-10 Homeodomain-Containing Homolog/Vsx2, Visual System Homeobox 2) is actively expressed in all neural retinal progenitor cells during development, cell-specific recombination occurs in all retinal neurons as well as supporting Müller glial cells. Figure 1b provides a simplified diagram showing *Chx10*-targeted retinal cell types (and their localization within the retina) as well as retinal cell populations not targeted by the NR-*Traf3* KO model.

To assess the effect of NR-*Traf3* depletion on whole retinal tissue TRAF3 protein expression, we performed Western blot analysis of retinal tissue extracts in adult NR-*Traf3* WT and KO mice (Figure 2). We found that NR-*Traf3* KO retinas exhibited a 74.6% reduction in total retinal TRAF3 protein compared to WT animals (Figure 2a). In a second cohort, we also observed a dose-dependent decrease in TRAF3 protein where heterozygous and homozygous NR-*Traf3* mice displayed 30.7% and 62.1% decreases, respectively, compared to WT controls (Appendix A). Thus, our data suggest that the neural retinal TRAF3 pool accounts for the majority of TRAF3 expression in the retina and that the NR-*Traf3* KO model provides a novel tool for assessing the importance of *Traf3* in the neural retinal (NR) compartment. To evaluate the specificity of *Chx10*-Cre-mediated recombination, we examined TRAF3 expression in brains of NR-*Traf3* KO mice and WT littermate controls and found no difference in TRAF3 expression (Figure 2b). Interestingly, we noted that the retina exhibited relatively higher TRAF3 levels compared to brain neuronal tissue (per μg total protein) as approximately 2.4× the amount of brain protein extract was required to detect TRAF3 within the linear range of the assay. We observed no differences in genotype-dependent body weights within male or female groups, and there were no sex- or genotype-dependent differences in brain or retinal tissue weights (Appendix A).

### 3.2. Neural Retinal Traf3 Depletion Does Not Adversely Affect Basal Retinal Structure or Function

Global *Traf3* depletion results in neonatal lethality around P10 in mice [19]. However, several viable non-ocular conditional *Traf3* KO models have been successfully generated [17,21,24]. To our knowledge, we have developed the first ocular model of conditional-*Traf3* depletion and have shown that TRAF3 protein is highly expressed in the NR compartment of retinal tissue (Figure 2). Thus, before investigating the potential role of TRAF3 in retinal immunity, we first wanted to identify whether NR-*Traf3* ablation had any impact on basal retinal tissue development, structure, and/or function prior to immune challenge.

#### 3.2.1. Retinal Tissue Structure and Patterning

To assess potential overall defects of NR-*Traf3* KO on retinal tissue development and/or morphology, we used in vivo spectral domain optical coherence technology/tomography (SD-OCT) to measure total retinal thickness (in eight distinct retinal zones) as defined by the distance from the inner limiting membrane of the retinal neural fiber layer (RNFL) to the outermost boundary of Verhoeff’s membrane located in the RPE (Figure 1 and Figure 3a).

There was no difference in total retinal thickness for any of the eight retinal zones when comparing NR-*Traf3* WT and KO retinas (Figure 3a). Additionally, we found no differences within either the smaller inner, middle, and outer retinal subdivisions (IRT, MRT, and ORT, respectively) or the individual sublayers that comprise these subdivisions of the retina (Appendix A). We also performed SD-OCT in NR-*Traf3* heterozygous (HET) animals (Appendix A). Interestingly, we did find small, but statistically significant differences between WT and HET INL of the inner nasal retinal zone IRL subdivision, which also affected the TRT. However, there were no statistical differences found for inner nasal retinal zone IRL RNFL or IPL sublayers individually. Thus, taken together, our SD-OCT results suggest that NR-*Traf3* depletion does not affect overall retinal structure, and that NR-*Traf3* KO mice exhibit normal retinal cellular arrangement and regular tissue configuration under basal conditions.

To validate that there was no overall effect of NR-*Traf3* depletion on retinal architecture, we analyzed H&E-stained adult retinal paraffin sections (Figure 3b). We observed normal lamination of all three neuronal layers (i.e., RGC, INL, and ONL), as well as intact RPE and choroidal vascular layers (Figure 3b and Appendix A). Vascular cross sections for all three retinal vascular layers were also present (Appendix A) and no edema was observed. Additionally, photoreceptor inner and outer segments could be visualized and appeared to have typical morphology. Müller glial development and differentiation also appeared normal as evidenced by immunofluorescent Glutamine Synthase (GS) detection in retinal paraffin sections (Appendix A).

#### 3.2.2. Visual and Retinal Function

To measure overall visual function, we used optokinetic tracking (OKT) to evaluate spatial vision as a measure of visual acuity. Using a virtual optomotor system, we measured the behavioral temporal-to-nasal optokinetic reflex in NR-*Traf3* WT and KO mice for each eye (left = LE, right = RE) [32]. We found no genotype-dependent differences in visual acuity for NR-*Traf3* KO animals compared to WT littermates (Figure 4; LE, WT: 0.397 ± 0.011, KO: 0.386 ± 0.012; RE WT: 0.398 ± 0.013, KO: 0.386 ± 0.029), suggesting that overall visual function was not impaired.

To measure retinal function, we used scotopic electroretinography (ERG) to assess NR-*Traf3* effects on rod photoreceptor-derived scotopic electrical stimuli for a range of flash intensities (Figure 5). We first generated response amplitude curves for both a- and b-wave responses (Figure 5a), which were then transformed into Vmax histograms for physiological interpretation (Figure 5b). We did not find any genotype-dependent differences in a-wave responses for any of the flash intensities tested, which corresponded to no difference in a-wave Vmax between groups (WT: 292.9 ± 23.4; KO: 264.8 ± 61.8). For b-wave intensity, we found a reduction in the NR-*Traf3* KO response at a flash intensity of −1 log cd × s/m^2^; however, this difference was not sufficient to alter the physiological b-wave Vmax compared to WT controls (WT: 602.2 ± 38.4; KO: 511.2 ± 103.2). Thus, we conclude that NR-*Traf3* does not significantly affect retinal function under basal conditions.

### 3.3. Neural Retinal Traf3 Promotes the Retinal Immune Response to Endotoxin-Mediated Inflammatory Activation

Intravitreal LPS injection (LPS^ivt^) has been established as an effective model for inducing acute murine ocular inflammatory leukocyte infiltration [39]. To initially assess whether NR-*Traf3* depletion affects retinal immune cell infiltration, we first performed a 3-color flow cytometry experiment on NR-*Traf3* WT and KO retinas 24 h after LPS^ivt^ induction (Appendix A). We observed an expected dramatic increase in the number of “Total Leukocytes” (CD45^+^ cells) from NR-*Traf3* WT retinas following LPS^ivt^ injection (66.2% increase; PBS: 2455 ± 512 vs. LPS: 7270 ± 1490; *p* = 0.001). Immune cell numbers also increased in NR-*Traf3* KOs with LPS^ivt^ treatment, though this did not statistically differ from that of contralateral PBS controls (37.5% increase; PBS: 1959 ± 525 vs. LPS: 3137 ± 604; *p* = 0.3571). Furthermore, LPS^ivt^-treated NR-*Traf3* KOs retinas exhibited significantly *decreased* infiltrate compared to their LPS^ivt^-treated NR-*Traf3* WT littermates (43.1% of WT response; NR-*Traf3* WT LPS: 7270 ± 1490 vs. KO LPS: 3137 ± 604; *p* < 0.01), suggesting that NR-*Traf3 promotes* retinal inflammation. Likewise, NR-*Traf3* KO retinas exhibited an LPS-dependent reduction in GR1^+^F4/80^−^ cells (WT PBS: 1055 ± 303, LPS: 4553 ± 765 vs. KO PBS: 640 ± 298, LPS: 1176 ± 413; *p* < 0.0001, LPS WT vs. KO), which accounted for the majority of total infiltrating leukocytes (CD45^+^ cells) and suggested that most pervading cells harbored a “Polymorphonuclear Leukocyte” (PMN) cell-type signature. While similar trends of reduced infiltrate were found for “Inflammatory Monocyte” (GR1^+^/F4/80^+^; WT PBS: 700 ± 209, LPS: 1439 ± 626 vs. KO PBS: 603 ± 187 LPS: 963 ± 275) and “Macrophage” (GR1^−^F4/80^+^; WT PBS: 214 ± 31, LPS: 402 ± 167 vs. KO PBS: 287 ± 87 LPS: 331 ± 76) monocytic/macrophage-like populations, these results did not reach statistical significance. Together, our initial 3-color flow cytometry data suggested that the primary cell population affected by NR-*Traf3* depletion was PMN cells.

To validate our findings, we subsequently performed 7-color flow cytometry on LPS^ivt^-treated NR-*Traf3* WT and KO retinas. In agreement with our initial results, we observed a considerable reduction in LPS-stimulated total myeloid infiltrate (Figure 6, population A, CD45^+^CD11b^+^) in NR-*Traf3* KO retinas (45% response compared to WT controls), where the majority of cells affected displayed a granulocytic/PMN signature (Figure 6, population D, CD45^+^CD11b^+^F4/80lowLy6G^+^Ly6C^+^). Using the more rigorous 7-color analysis, we also found significant genotype-dependent differences in LPS-stimulated monocyte/macrophage populations (Figure 6, population B, CD45^+^CD11b^+^F4/80^+^ CD115^+^CCR2^+^Ly6C^+^; and population C, CD45^+^CD11b^+^F4/80^+^ CD115^+^CCR2^−^Ly6C^+^), although total cell numbers for these populations were much lower than for the granulocyte population (Figure 6, population D). Thus, our findings support that NR-Traf3 promotes retinal immune cell infiltration following 24 h LPSivt immune activation.

## 4. Discussion

Inflammatory diseases of the CNS are not well understood and are further confounded by the unique immune-privileged environment displayed by neuronal tissues. Thus, for the CNS in particular, molecular targeting of novel inflammatory regulators with neuronal-specific functions is an attractive strategy for treatment of inflammation-associated neurodegenerative diseases. Due to the greater accessibility of retinal tissue to laboratory manipulations and in vivo assessments, retinal mouse models are frequently used to study retinal inflammatory disease and are considered a precursory conduit for understanding neuronal inflammatory conditions of the brain. It is well known that microglia, astrocytes and the retinal vasculature play important roles in retinal immune response regulation [12,40,41]. Previously, we found that the NR compartment also plays an important role in retinal immunity [13]. The NR houses all retinal neurons as well as Müller glial cells, and accounts for approximately 98.9% of total cells that comprise retinal tissue in healthy adult mice (Figure 1b) [42]. In our previous study, we found that targeted NR depletion of the *Caveolin-1* (*Cav1*) gene in the NR was sufficient to blunt LPS-induced retinal immune cell infiltration, suggesting that the NR compartment, specifically, plays an important role in retinal immune regulation [13]. Interestingly, we also observed upregulation of the immune modulator TRAF3 in retinal tissue membrane fractions from NR-*Cav1* mice. As the role of TRAF3 in the retina has not been investigated, we have generated a NR-*Traf3* depletion model to enable examination of retina-specific TRAF3 functions.

At present, there have been no studies conducted on retinal-*Traf3* function. However, gene and protein databases have reported high levels of TRAF3 expression in the retina [27]. Here, we have confirmed that TRAF3 protein is highly expressed in healthy murine retinal tissue, and that the *Chx10*-Cre model specifically targets retinal, but not brain, neuronal tissue TRAF3 (Figure 2). As the NR contains numerous neuronal subpopulations (as well as Müller glia), we attempted to identify retinal cell type-specific TRAF3 expression via immunofluorescence using several antibody/tissue preparation conditions. However, of all TRAF3 antibodies tested that produced an immunofluorescence signal, none were able to demonstrate a genotype-dependent reduction in immunofluorescence staining that would provide confident identification of neuronal subtype-specific TRAF3 expression. Interestingly, although our Western blot data showed a significant NR-*Traf3* KO-dependent reduction in relative TRAF3 protein, we noted that two sets of high-molecular weight non-specific bands were also detected with this antibody (Appendix A). Thus, while this reagent was able to clearly demonstrate depletion of TRAF3 in our whole retinal lysate samples, future users should use caution and ensure sufficient gel separation of TRAF3 protein from other proteins near the expected molecular weight of 65 kDa that interact with this antibody. While the difficulty in TRAF3 detection via tissue immunofluorescence may be in part due to non-specific binding of this reagent, we also tested various additional antibodies in conjunction with varied antigen-retrieval techniques. We were unable to confidently and specifically identify TRAF3 protein retinas using immunofluorescence, which may require further modification of tissue fixation and antigen-retrieval procedures and/or development of reagents with greater specificity toward retinal TRAF3 protein. Nevertheless, our data support that the NR-*Traf3* KO model specifically targets retinal tissue and that the majority of retinal TRAF3 protein expression resides within the NR compartment (Figure 2). This agrees with previous studies that reported antibody detection of TRAF3 in rat spinal cord neurons, mouse brain neurons, and human cerebral and Purkinje neurons, but not astrocytes or oligodendroglia [43,44]. Interestingly, Krajewski, et al. also reported that the level of TRAF3 expression varied among different neuronal subpopulations of the brain [44]. Thus, it is likely that the high NR TRAF3 expression observed in our study is largely due to neuronal-derived expression; however, the level of expression may vary among the various retinal neuronal subtypes and Müller glial-derived TRAF3 expression cannot be excluded. Future studies aim to elucidate retinal cell-specific TRAF3 protein detection via further development of immunofluorescence methods, as well as using alternative flow cytometry and cell culture techniques.

Numerous non-retinal TRAF3 studies have demonstrated a breadth of Traf3 functions, including modulation of cell survival, metabolic, and immune processes [15,18,22,45,46]. Given the complex cellular composition and high metabolic demands of retinal tissue, we first characterized the impact of NR-*Traf3* depletion on retinal structure and function to determine whether our new NR-*Traf3* model was suitable for further manipulations that would allow for reliable assessments of TRAF3 function. Using our in-depth in vivo SD-OCT and histological analyses, we found no apparent overall genotype-dependent defects on retinal tissue architecture, lamination, or patterning (Figure 3 and Appendix A). Interestingly, we did identify differences in the inner nasal zone INL of NR-*Traf3* heterozygous animals, which statistically affected the TRT for this retinal zone in HET mice. However, all data considered, we believe this singular difference was likely due to marginal biological variability rather than an indicator of meaningful physiological variances as no statistical differences were found for the INL of other retinal zones, or when comparing thicknesses of any of the other sublayers. Moreover, we observed no differences in tissue morphology by H&E staining (Appendix A). Furthermore, we found that NR-*Traf3* depletion did not significantly alter retinal electrophysiology or spatial vision (Figure 4 and Figure 5). Taken together, our data suggest that *Traf3* deletion in the NR compartment does not result in developmental or degenerative abnormalities and that the NR-*Traf3* mouse is a suitable model for investigations of NR-derived TRAF3 function.

TRAF3 is ubiquitously expressed and global-*Traf3* deletion results in neonatal lethality in mice, which is likely due to systemic effects on the immune system that result in a hyperinflammatory state [19,44]. In fact, the majority of TRAF3 studies are focused on TRAF3-regulated immune cell-specific functions. Collectively, these studies identified TRAF3 as a cytosolic and nuclear adaptor protein that regulates numerous cell surface receptors and downstream effectors in a highly context- and cell-specific manner in order to regulate various cellular processes involved in immune cell development (natural killer T cells), survival (B cells), and metabolism (myeloid and B cells) [17,18,47]. Additionally, TRAF3 can either positively or negatively modulate inflammatory and anti-viral signaling pathways, which is cell type- and receptor-dependent [47,48,49,50]. For an extensive overview on the complexities of Traf3 functions in immune cells, the reader is referred to previously published reviews [18,45,47].

In recent years, numerous studies have utilized conditional Traf3 knockout models to reveal additional roles for non-immune cell, tissue-specific TRAF3 functions. For example, TRAF3 is expressed in both osteoclasts and osteoblasts, and thus plays important roles in both bone resorption and formation during skeletal development and bone remodeling, respectively [15,51,52,53]. Here, our data show that NR-*Traf3* promotes the retinal immune response by facilitating LPS-induced inflammatory immune cell infiltration (Figure 6 and Appendix A). This agrees with previous data showing that conditional-*Traf3* depletion in the liver results in reduced immune infiltration following ischemia/reperfusion injury, whereas hepatic *Traf3* overexpression exacerbates the level of infiltrate [54]. In this study, canonical NFκB signaling pathway effectors and inflammatory cytokine gene expression targets were shown to be similarly upregulated by liver-derived TRAF3 and were assumed to be responsible for the effects on hepatic tissue inflammatory infiltrate. Importantly, another study using conditional depletion of *Traf3* in brain-specific neurons showed that TRAF3 also activates canonical inflammatory NFκB signaling in the brain [24]. Thus, it is probable that a primary function of retinal TRAF3 is to regulate retinal immune responses. However, it is conceivable that NR-*Traf3* harbors the ability to modulate additional retinal signaling pathways and cellular processes as TRAF3 has been shown to directly and indirectly interact with multiple immune receptors, respond to a variety of upstream stimuli, and regulate molecular effectors belonging to diverse signaling pathways. In the heart, TRAF3 mediates GPCR (G-coupled protein receptor)-stimulated cardiac hypertrophy through activation of TBK1-AKT-mTOR signaling, which results in increased protein synthesis that contributes to the hypertrophic response [23]. In the liver, TRAF3 contributes to hepatic cell death following ischemia/reperfusion injury and to hepatic steatosis following HFD (high-fat diet)-induced diabetes, in part, through promotion of via TAK1-MKK-JNK signaling [21,54]. Likewise, in the brain, TRAF3 promotes neuronal cell death partly through TAK1-MKK-JNK activation in both ischemia/reperfusion and subarachnoid hemorrhage injury models [24,28]. Zhang et al. also showed that TRAF3 promoted neuronal apoptosis following spinal cord injury via thoracic vertebral contusion; however, whether this was also dependent on TAK1-MKK-JNK activation was not investigated in this study [29]. Here, we show that NR-TRAF3 regulates TLR4-specific immune activation by LPS endotoxin administration. Thus, as our previous work in NR-*Cav1* KO animals also showed blunted immune cell infiltration, the simultaneous TRAF3 upregulation we observed might reflect the retina’s attempt to overcome the inability of NR-*Cav1* to promote the immune response (rather than suggest that TRAF3 is responsible for the blunted response) [13]. Our future studies include identification of NR-TRAF3-dependent signaling pathways involved in the retinal immune response to various inflammatory stimuli, as well as developing a more thorough investigation of TRAF3-dependent infiltrating populations, including immune cells of both lymphoid and myeloid lineages. Additionally, we plan to investigate corresponding retinal cytokine/chemokine responses to various inflammatory stimuli, as well as potential interactions between CAV1 and TRAF3. Our novel NR-Traf3 model can also be used to investigate potential non-immune retinal TRAF3 functions.

## 5. Conclusions

To our knowledge, we have generated the first retinal-specific conditional *Traf3* knockout mouse model (NR-*Traf3* KO) using *Chx10*-Cre-mediated targeting of the neural retinal (NR) compartment. Our analyses of retinal tissue from NR-*Traf3* KO animals suggest that this model provides a valuable tool for investigating neuronal Traf3 functions. Here, we also provide preliminary evidence that NR-*Traf3* regulates the retinal immune response as NR-*Traf3* ablation was sufficient to blunt LPS-mediated retinal immune cell infiltration. While the precise mechanism by which NR-*Traf3* promotes immune infiltration needs further investigation, our study agrees with existing literature that suggests therapeutic targeting of neuronal *Traf3* may provide neuroprotection and prevent chronic inflammation that occurs with neurodegenerative diseases of the CNS.

## Figures and Tables

**Figure 1 cells-10-02068-f001:**
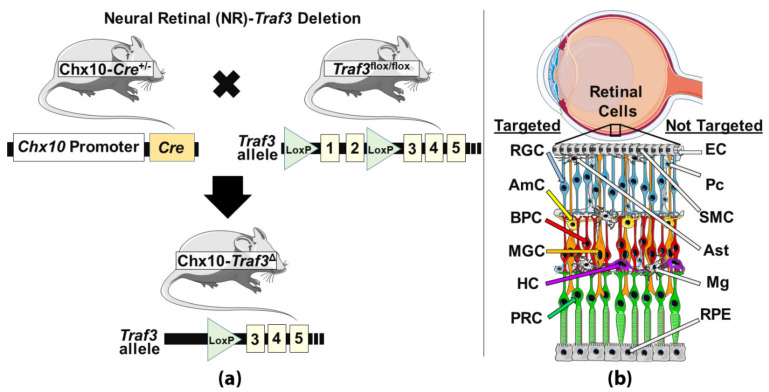
Schematic diagram showing neural retinal (NR)-specific genetic deletion of *Traf3* in the *Chx10*-Cre/Traf3^flox/flox^ (NR-*Traf3* KO) mouse model. (**a**) Breeding strategy for NR-specific *Traf3* depletion. (**b**) Retinal tissue diagram outlining resident retinal cell populations targeted (and not targeted) for *Traf3* deletion using the NR-*Traf3* KO model. Targeted: RGC, retinal ganglion cell; AmC, amacrine cell; BPC, bipolar cell; MGC, Müller glial cell; HC, horizontal cell; PRC, photoreceptor cell. Not targeted: EC, endothelial cell; Pc, pericyte; SMC, smooth muscle cell; Ast, astrocyte; Mg, microglia; RPE, retinal pigment epithelia.

**Figure 2 cells-10-02068-f002:**
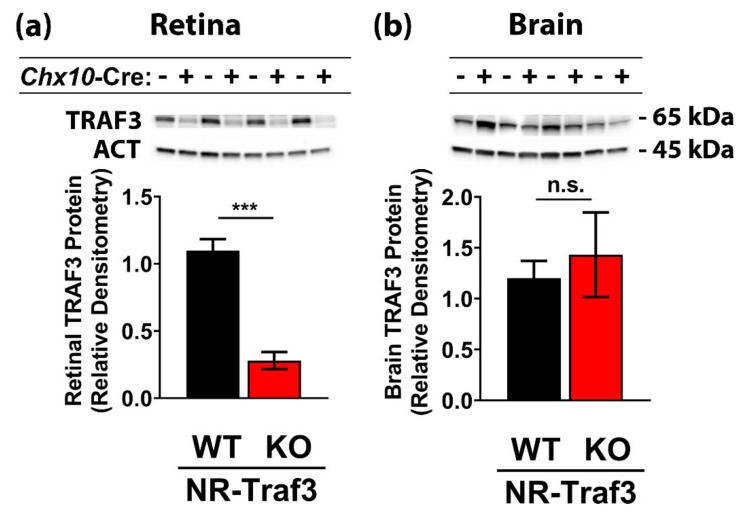
NR-*Traf3* KO mice exhibit significant and specific reductions in whole retinal tissue TRAF3 protein. TRAF3 Western blot data and corresponding quantitative densitometry analysis for neuronal whole retinal (**a**) and brain (**b**) tissues extracts, respectively. β-actin was used as loading control and for normalization of TRAF3 protein expression. To detect protein within the linear range of the assay, protein loads were as follows: retinal TRAF3, 25 µg; brain TRAF3, 60 µg; retinal/brain β-actin, 10 µg. *Chx10*-Cre negative (−) and positive (+) expression indicate NR-*Traf3* WT and KO mice, respectively. Data are the mean ± SEM and were analyzed via Student’s *t*-test (*** *p* < 0.001; *n* = 4).

**Figure 3 cells-10-02068-f003:**
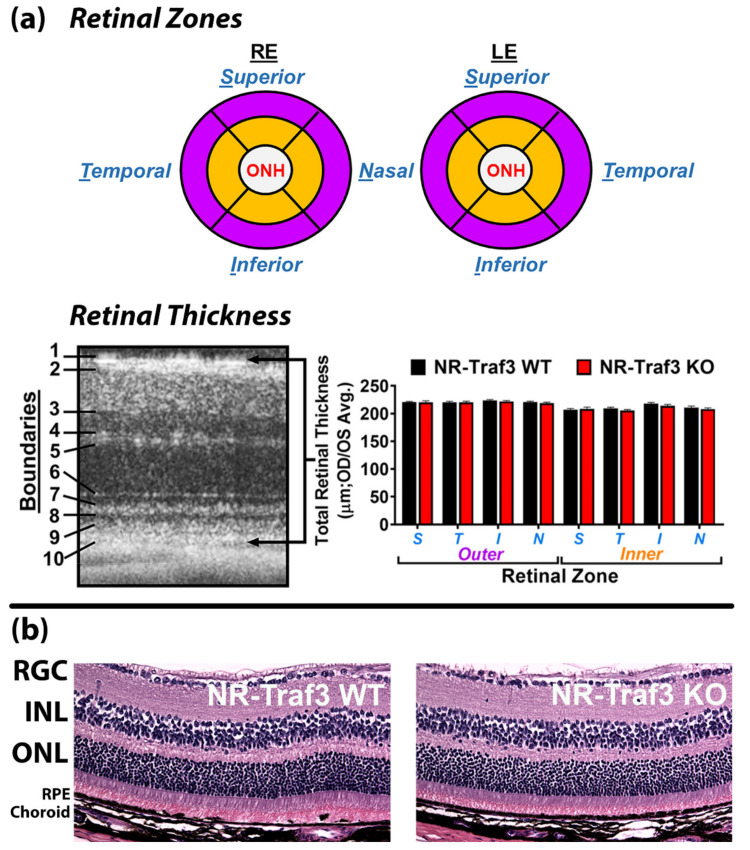
NR-*Traf3* KO mice display normal retinal structure and lamination. (**a**) Diagram showing distinction of outer (purple) and inner (orange) retinal regions with further separation into superior, temporal, inferior, and nasal zones. Total retinal thickness (TRT) was measured for all 8 retinal zones for both eyes and was defined as the total distance (in µm) between boundaries 1 (inner limiting membrane) and 10 (Verhoeff’s membrane), which is illustrated in the example SD-OCT image. Additional boundaries used to measure thicknesses of subdivisions and individual sublayers within the TRT are also noted in the example SD-OCT image (see also Appendix A). Numerical data for TRT and retinal sublayers are provided in Appendix A. (**b**) representative H&E-stained retinal sections showing normal lamination of retinal layers: RE, right eye; LE, left eye; ONH, optic nerve head; S, superior; T, temporal; I, inferior; N, nasal; RGC, retinal ganglion cell layer; INL, inner nuclear layer; ONL, outer nuclear layer; RPE, retinal pigment epithelial layer. TRT data are represented as the mean ± SEM. (*n* = 6; Student’s *t*-tests for each retinal zone were not significant).

**Figure 4 cells-10-02068-f004:**
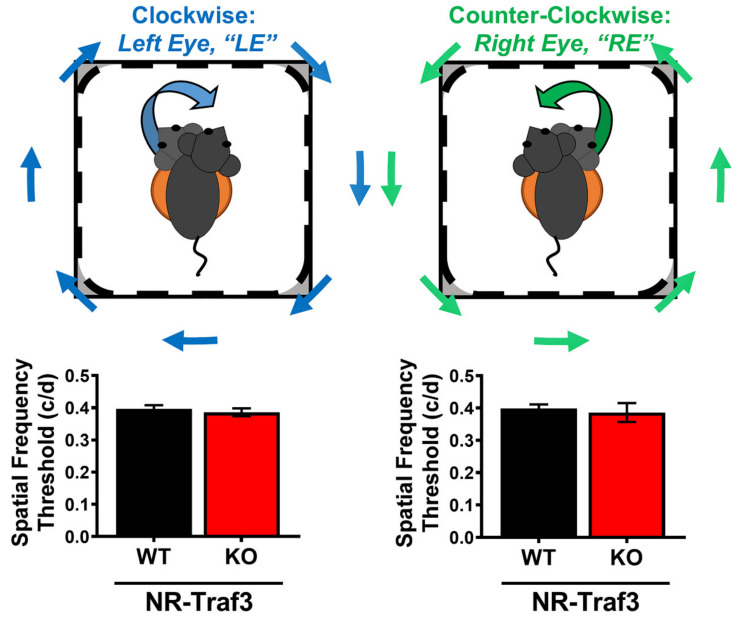
NR-*Traf3* depletion does not affect basal visual acuity. Schematic (top) shows virtual cylinder grating rotation direction for left eye (LE, clockwise) and right eye (RE, counter-clockwise) spatial frequency determinations. Visual acuity data for left and right eyes show no difference in visual thresholds between NR-*Traf3* WT and KO animals. (*n* = 5–12; Student’s *t*-test, not significant).

**Figure 5 cells-10-02068-f005:**
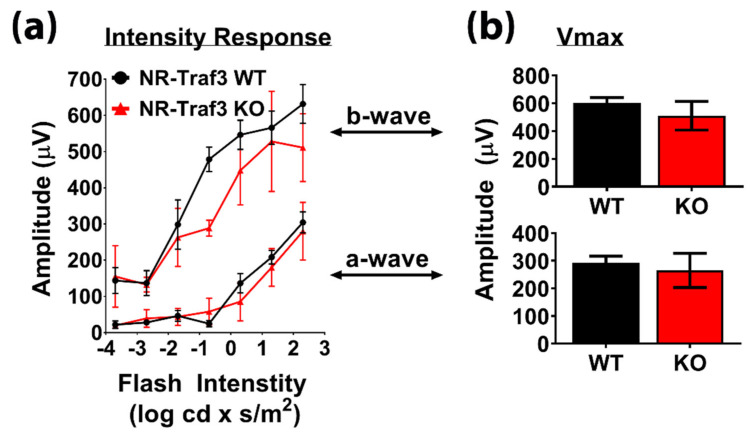
NR-*Traf3* KO mice exhibit normal basal retinal function. Scotopic electroretinography (ERG) data represent electrical response curves for (**a**) increasing flash intensity exposures. Response curves were used to determine (**b**) maximum amplitudes for rod photoreceptor-derived a-wave and b-wave responses. NR-*Traf3* KO resulted in decreased b-wave amplitude at −1 log cd × s/m^2^; however, there was no physiological difference in Vmax for either a- or b-waves between NR-*Traf3* WTs and Kos. (*n* = 3–7; Student’s *t*-test, not significant).

**Figure 6 cells-10-02068-f006:**
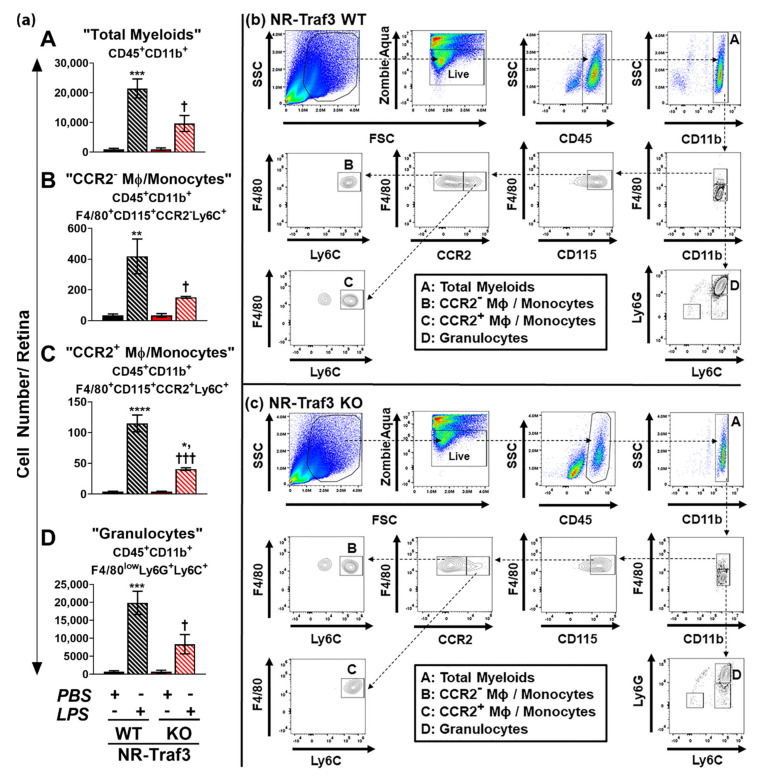
NR-*Traf3* promotes LPS-induced retinal immune cell infiltration (7-color). Flow cytometry data representing (**a**) grouped histogram quantification and (**b**) representative NR-*Traf3* WT and (**c**) KO dot plots for 7-color flow cytometry detection of immune cells from whole retinal single cell suspensions following 24 h intravitreal LPS administration. Data for (**a**) are the mean ± SEM and were analyzed via two-way ANOVA with Tukey’s multiple comparisons post hoc test (LPS effect: **** *p* < 0.0001, *** *p* < 0.001, ** *p* < 0.01, * *p* < 0.05; Genotype effect: ^†††^
*p* < 0.001, ^†^
*p* < 0.05; *n* = 3). Immune cell populations of interest are designated A–D: A, total myeloids (CD45^+^CD11b^+^); B, CCR2^−^ macrophages (MΦ)/monocytes (CD45^+^CD11b^+^F4/80^+^ CD115^+^CCR2^+^Ly6C^+^); C, CCR2^+^ macrophages (MΦ)/monocytes (CD45^+^CD11b^+^F4/80^+^ CD115^+^CCR2^−^Ly6C^+^); D, granulocytes (CD45^+^CD11b^+^F4/80^low^Ly6G^+^Ly6C^+^).

## Data Availability

The data supporting the findings of this study are available from the corresponding author upon reasonable request.

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
