# Peer review of "The Chx10-Traf3 Knockout Mouse as a Viable Model to Study Neuronal Immune Regulation"

_cells, 2021, doi:10.3390/cells10082068_

Round 1

Reviewer 1 Report

In this work, the effects that NR-Traf3 depletion has on the expression of the TRAF3 protein on whole retina, visual acuity, and the function and structure of the retina, as well as its regulatory role in retinal immunity, are studied. The authors show that, although this protein is highly expressed in the retina, its depletion does not alter the retinal function/structure, but it does alter the immune infiltration in the retina, for which the authors propose that it acts as a regulator of immunity in the retina. This work provides novel and solid evidences in this regard, which is why it deserves to be published. However, there are several questions that must be answered previously.

Main points

Certain results should be explained in the discussion, for example, if TRAF-3 protein is found in brain and retina, what could be the reason why the depletion caused by NR-Traf3 KO (which is expected to be general) only leads to a significant decrease of said TRAF3 protein in the retina, but not in the brain?

It is a pity that in the study of visual acuity by optokinetic tracking they only measure the spatial frequency threshold. More complete information could be given by showing the response to different spatial frequencies.

Since it is possible that different cells of the neural compartment of the retina have different expression of TRAF3 protein, it would be interesting to make multiple antibody labels to detect the TRAF3 protein together with specific antibodies against these cells (Protein kinase C for bipolar cells, calbindin for horizontal cells, brn3a for ganglion cells, etc.). This information, in addition to quantifying the expression of TRAF3 in the different cells of the retina, could help to determine the reason why significant differences are observed in the inner nuclear layer between both mouse models. 

Minor points

Immunohistochemistry should be described in more detail. The precautions taken at the time of intraocular injection to avoid unwanted damage must be declared also.

The Graphpad Prism version of the statistical program should be provided.

In the methodology, it must be declared what standards for the care of laboratory animals have been met. The use of the word "euthanized" is not appropriate, it would be better to use the word "killed" or "sacrificed".

Why do the authors use the acronyms OD and OS, instead of RE and LE?Figure 5 should be separated into two sub-figures, a) showing the amplitude / intensity curve and b) the quantification of the maximum response for both types of waves. On the other hand, the graphs of such quantization are mistaken; the superior corresponds to a wave (instead of b wave) and the inferior one to b wave (instead of a wave). Likewise, the units should be indicated in these graphs.

Reviewer 2 Report

Present manuscript describes interesting work on the development of mice model of Chx10-Traf3 Knockout Mouse as a Viable Model to Study Neuronal Immune Regulation. The work is extremely well written and I think can be published as it is. Just few minor suggestions.

  1. Introduction please avoid put them in form of subsection
  2. Schematic is confusing. The works talks about development of mice model studying the immune regulation. No information is provided in this regard.
  3. Supplementary figure S1 A., was there any statistical analysis done?
